

# Using genetic programming to predict and optimize protein function

Iliya Miralavy[1,2], Alexander R. Bricco[1,3], Assaf A. Gilad[1,4] and Wolfgang Banzhaf[1,2,5]

[1] BEACON Center of Evolution in Action, Michigan State University, East Lansing, MI, United States of America

[2] Department of Computer Science and Engineering, Michigan State University, East Lansing, MI, United States of America

[3] Department of Biomedical Engineering, Michigan State University, East Lansing, MI, United States of America

[4] Department of Chemical Engineering, Michigan State University, East Lansing, MI, United States of America

[5] Department of Computer Science, Memorial University of Newfoundland, St. John's, Newfoundland and Labrador, Canada

## ABSTRACT

Protein engineers conventionally use tools such as Directed Evolution to find new proteins with better functionalities and traits. More recently, computational techniques and especially machine learning approaches have been recruited to assist Directed Evolution, showing promising results. In this article, we propose POET, a computational Genetic Programming tool based on evolutionary computation methods to enhance screening and mutagenesis in Directed Evolution and help protein engineers to find proteins that have better functionality. As a proof-of-concept, we use peptides that generate MRI contrast detected by the Chemical Exchange Saturation Transfer contrast mechanism. The evolutionary methods used in POET are described, and the performance of POET in different epochs of our experiments with Chemical Exchange Saturation Transfer contrast are studied. Our results indicate that a computational modeling tool like POET can help to find peptides with 400% better functionality than used before.

Corresponding authors
Assaf A. Gilad, gilad@msu.edu
Wolfgang Banzhaf,
banzhafw@msu.edu,
banzhaf@mun.ca

# INTRODUCTION

Advances in computational techniques for learning and optimization have been extremely helpful in peptide (sequences of amino acids) design and protein engineering. Proteins are the workhorses of life, the machinery and active components necessary for allowing biological organisms to survive in their environment. Throughout millions of years, biological evolution has found a vast variety of proteins. Literature suggests $\sim 20,000$ non-modified proteins have been discovered in human body so far following the hypothesis that one gene produces one protein (*Ponomarenko et al., 2016*). However, not all of the protein search space has been explored by natural evolution yet. In science, protein structure and function prediction has been an active topic for structural biology and computer science. Protein engineers mainly use the three methods of Rational Design, Directed Evolution (DE) and *De Novo* design to find their proteins of interest. Rational Design creates new

molecules based on extensive prior knowledge of the 3D structure of proteins and their properties. Directed Evolution does not require as much prior knowledge for protein optimization and instead, uses random mutagenesis and screening of variants to find fitter proteins. *De novo* design uses computational means to design algorithms that learn from the 3D protein structure and their folding mechanisms to synthesise novel proteins (*Singh et al., 2017*).

Computational approaches to the design of proteins have been examined since the early 1980s (*Hellinga, 1998*). In that line of work, linear sequences of amino acids representing proteins in their basic sequential information are given to algorithms as inputs in order to create models able to predict their secondary properties and to predict features of their variants, such as inter-residue distances, 3D structural shapes, and protein folding. Obtaining this information is crucial to predicting protein function and creating new variants. Different techniques have been applied to this problem to date, some of which formulate it as a classification problem where a classifying algorithms aims to find the protein class with respect to structural properties or traits of the peptide under consideration. Others have taken a more continuous approach and predict numerical values corresponding to the characteristics of a protein structure. In that case, the applied techniques are trained to estimate unknown numerical properties of a protein, like inter-residue distances or hydrophobicity levels.

## Protein engineering by Directed Evolution

As we said before, proteins and peptides play an extremely important and integral part in the life cycle of organisms. They carry out all kinds of natural functions such as forming muscle tissues, creating enzymes and hormones, and are the building blocks of food and bio-medicine. Looking at a living cell as a factory would make proteins the workers. These complex structures are formed from sequences of amino acids, which code for their structural and phenotypic properties (*Alberts et al., 2017*). Evolution by natural selection has produced numerous protein wild-types through millions of years, yet has only explored a fraction of the vast protein search space. There are a total of 20 amino acids that can code for proteins. To find a peptide consisting of 10 amino acids in a search space of $20^{10}$ amino acid sequences is a complex undertaking. Exploring this large search space in order to discover new and possibly better protein variants is one of the activities of protein engineering.

As protein engineers studied proteins to understand these complex structures better and optimize their functionalities or even develop new protein functions, they came up with DE, now a common technique for reaching these goals (*Arnold, 1998*). The DE technique starts with a pool of proteins with similar functionalities to the desired one and imitates mutation and natural selection in order to create the next generation of fitter proteins, ultimately optimizing them. Since DE is performed *in vitro*, it is a time-consuming and costly approach that requires careful monitoring and screening of massive numbers of mutant proteins in each generation.

In recent years, many researchers have started to use computational and Machine Learning (ML) methods to generate models that can predict the phenotypic behavior

of proteins, based on their genetic and molecular makeup (*Yang, Wu & Arnold, 2019*). In contrast to wet lab experiments, however, using computational models enables the exploration of more of the search space in a significantly shorter amount of time. These computational models need to be well-trained in order to be effective which requires a lot of data. Figure 1 shows a general overview of the difference between a conventional Directed Evolution and a Machine Learning-guided Directed Evolution (ML-DE). In DE, after monitoring, unimproved mutants are discarded, and only improved proteins get the chance to be selected for diversity generation. Unfortunately, this approach often causes DE to get stuck in local optima. Meanwhile, in ML-DE, computational models choose potential mutants allowing to cover more of the search space and thereby increasing the chance of escaping from local optima. Even though the state of the art has not reached this point, an ideal computational model should accurately predict any protein function in a matter of seconds greatly reducing the need for wet lab methods such as DE; it could therefore significantly benefit the world of science and medicine.

## Genetic programming

Genetic Programming (GP) (*Koza, 1992*; *Banzhaf et al., 1998*) is an extension of the Genetic Algorithm (GA) (*Holland, 1992*), an algorithm inspired by the Darwinian conception of natural evolution (*Darwin, 1909*), in which computational problem-solving models—usually in the form of computer programs—are evolved and optimized over a repeated generational cycle. Although all GP algorithms follow a core inspired by natural evolution, they come in various forms and representations. Normally, the GP process starts with a population of random individuals representing unknown solutions to a given problem. These individuals are evaluated by a pre-defined fitness function to measure how well they can solve the problem. A selection mechanism is then used to choose parent individuals that will undergo evolutionary operators (crossover and mutation) and form the next generation of population. Usually, fitter individuals have a better chance to get selected. During crossover, parts of the representation of the selected parent individuals are combined to create one or more offspring individuals with details depending on the algorithm. The mutation operator randomly alters parts of the newly created offspring individuals. Sometimes, an additional survival selection step filters out only the better offspring for inclusion in the population. The entire cyclical process continues until a termination condition, such as finding a model that satisfies user requirements, is met.

## Protein optimization engineering tool

Some of the design principles in developing protein-function-predicting models include determining (i) how much and what type of information should be given to the system, (ii) to what extent the results should be trusted, and finally, (iii) what type of problem solver should be used. We propose Protein Optimization Engineering Tool (POET), a GP computational tool that can aid DE in order to find new proteins with better functionalities. The magnetic susceptibility or Chemical Exchange Saturation Transfer (CEST) contrast of peptides is our goal here to show a proof-of-concept of the efficacy of the method.

CEST (*van Zijl & Yadav, 2011*) is a magnetic resonance imaging (MRI) contrast approach in which peptides with exchangeable protons or molecules are saturated and

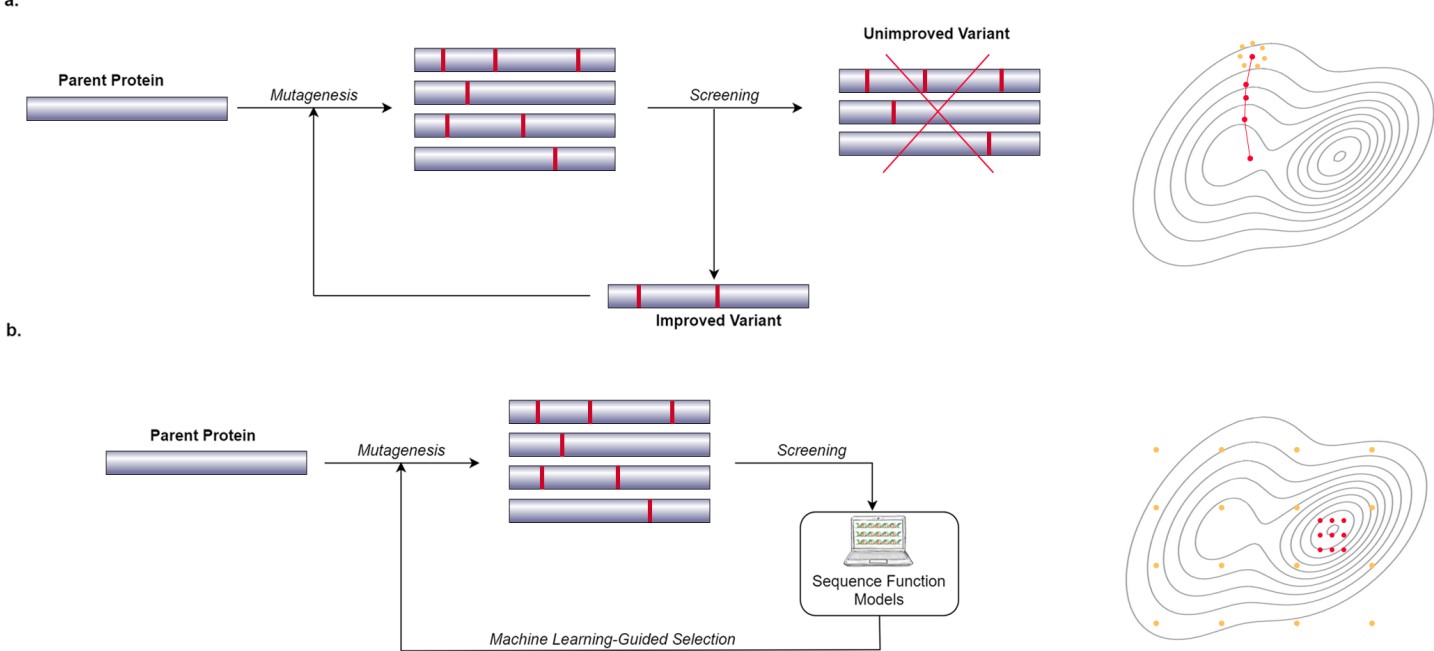

**Figure 1** **Difference between conventional DE and ML-DE.** (A) Conventional DE starts with parent protein undergoing mutagenesis to produce close variants. Detailed lab screening is done to throw out the unimproved variants and select the improved ones for the next generation of the mutagenesis. This process continues until a desirable mutant is found. (B) Commonly in ML-DE, sequence-function models replace the rigorous screening and selection task in conventional DE. ML-DE is able to explore more of the search space in the same amount of time and has a lower risk to get stuck in a local optimum. In (B) any of the orange dots can be rationally chosen as the start of the optimization, while in (A) the search space is limited to the direction of the search.

detected indirectly through enhanced water signals after transfer. The main advantage of CEST based proteins is that they can be encoded into DNA and expressed in live cells and tissue, thus allowing tracking non-invasively with MRI (*Gilad et al., 2022*; *Airan et al., 2012*; *Gilad et al., 2007*; *Perlman et al., 2020*). POET here aims to aid DE by predicting better functioning CEST contrast proteins and replacing the rigorous and costly task of screening and mutagenesis.

Figure 2 gives a high-level overview of the POET structure. POET can be divided into the three major phases of (i) model training, (ii) protein optimization & prediction, and (iii) wet lab experiments. During model training, POET receives an abstract dataset of amino acid sequences in FASTA format and a trait value corresponding here to their CEST contrast. POET uses GP to learn valuable motifs in the protein sequences, assigns weights to them, and creates collections of rules (motifs and weights) as its models. Given a pool of protein sequences to choose from, an optimized POET model can then select potential proteins with regard to the best expected CEST behavior. During this protein optimization & prediction phase, a set of random protein sequences of arbitrary length (here set to 12) are chosen to go through mutagenesis to find highly fit sequences. Evaluation is performed using the best previously trained POET model. Once enough generations of mutation and selection have been done, POET chooses the proteins that are fittest in predicted function

**Figure 2** **A high-level overview of POET.** POET starts with training its sequence-function models using a curated dataset of protein sequences and their respective CEST contrast values (Model Training). Protein Optimization evolves a random pool of protein sequences to predict fitter variants by using previously trained models for evaluation. Different colors for protein sequences indicate that POET can start from a random pool and is able to find fitter proteins in a different search space from the starting pool. Predicted variants are evaluated in wet labs, and their measured values are added back to the protein dataset to evolve fitter CEST predicting models. The small computer sign on the left side of the proteins differentiates between natural and computational protein sequences.

for the third phase, wet-lab experiments. During wet-lab experiments, the predicted sequences are chemically synthesized, and their respective CEST contrast is measured in MRI. This concludes one round of POET which we call an epoch. The measured values are added to the protein dataset and the POET experiment continues for another round. The addition of newly added data is expected to improve the dataset and the potency of POET models to predict fitter sequences in the next epoch. POET omits the limitation of the costly and time-consuming monitoring of the peptides in each generation of DE and predicts a set of proteins that show potential based on the previous results found in its already known dataset.

The rest of the article is structured as follows: Section 2 discusses the related literature, Section 3 introduces materials and methods used for developing POET. In Section 4 reports on our empirical findings and results. Section 5 summarizes and gives a brief perspective on next steps and possible future research directions.

## RELATED WORKS

The scientific literature indicates how practical computational approaches can be in protein engineering and particularly ML-DE. In the late 1990s, Simulated Annealing (SA) was adopted to predict and discover the structure of a hyperthermophile variant of a protein that could easily bind to human proteins (*Malakauskas & Mayo, 1998*). They used Streptococcal protein $G\beta1$ domain ($G\beta1$) as the wild-type protein and managed to find a more stable variant with a melting temperature higher than 100 °C. In 2005, *Sim, Kim & Lee (2005)* used a fuzzy K-Nearest Neighbor (KNN) algorithm to predict how accessible protein

residues are to solvent molecules. They incorporated PSI-BLAST profiles as feature vectors and showed accuracy improvements in comparison to neural networks and support vector machines. They used a reference dataset of 3460 proteins and a test dataset of 229 proteins to evaluate their models. *Wagner et al. (2005)* tackled the same problem but by employing linear support vector regression and showed the applicability of such computationally less expensive method in predicting protein folding and relative solvent accessibility of amino acid residues. In 2020, *Xu et al. (2020)* compared the accuracy performance of 44 different ML techniques on four public and four proprietary datasets, showing that different datasets cause dissimilar performance levels for the same algorithms; nonetheless, most ML techniques show good promise in the area. Notably, *Xu (2019)* used deep learning to predict inter-residue distance distribution and folding of protein variants with fewer homologs (protein with the same ancestry) than commonly required. DeepMind (*Senior et al., 2020*) introduced AlphaFold, which applies deep residual-convolutional networks with dilation to predict protein inter-residue distances and benefited from this intermediary data to accurately predict protein shapes with a minimum TM-Score[1] of 0.7 on 24 out of 43 free modeling domains outperforming the previous best methods. In 2021, AlphaFold2 was released (*Jumper et al., 2021*) that can predict the 3D structure of proteins greatly close to the experimental results from their sequence representation outperforming its previous version. AlphaFold2 uses a different neural network system than its predecessor. The key differences in the neural network system is the allowance of continuous refinement of all parts of the structure, a novel equivariant transformer and a loss term that emphasizes on the orientational correctness of the residues. Most related researches in the field of protein engineering focus on predicting the structural properties of the proteins. However, the presented work aims to predict the phenotypic performance of the proteins from their sequence representation by evolving sequence-function models.

Many research groups have shown that evolutionary approaches can be valuable in solving the protein structure and function prediction challenge. *Siqueira & Venske (2021)* defines the Protein Structure Problem (PSP) and introduces different classes of evolutionary algorithms and quality metrics to solve the problem. In 1997, *Khimasia & Coveney (1997)* defined this challenge as an NP-Hard optimization problem and, aside from performance evaluation, showed how a Simple Genetic Algorithm (SGA) can be applied to evolve simple lattice-based structure-predicting models. *Rashid et al. (2012)* examined five variants of the GA for solving a simplified protein structure prediction and applied three methods of (i) exhaustive structure generation, (ii) hydrophobic-core directed macro-move, and (iii) a random stagnation recovery for enhancing each of these algorithms. *Koza & Andre (1999)* used GP and the idea of Automatically Defined Functions (ADFs) to predict the family of D-E-A-D box proteins. UniRep (*Alley et al., 2019*) applied sequence-based deep representation learning to generate an evolutionary, semantically, and structurally rich representation of protein properties that heuristic models could incorporate to predict structures and functions of unseen proteins. *Seehuus, Tveit & Edsberg (2005)* and *Seehuus (2005)* utilized Linear Genetic Programming (LGP) (*Brameier & Banzhaf, 2007*) for motif discovery. Their results indicated a better accuracy for discovering motifs in different protein families than traditional tree GP. *Fathi & Sadeghi (2018)* proposed a GP approach

[1]TM-Score is a metric to quantify similarities between topological structures of proteins. Scores lower than 0.17 correspond to unrelated proteins, while scores higher than 0.5 generally indicate the same fold.

to predict peptide sequence cleavage by HIV protease, and *Langdon, Petke & Lorenz (2018)* used Grow and Graft GP (GGGP) to predict the RNA folding of molecules. *Borro et al. (2006)* employed Bayesian classification to extract features from protein structure and achieves an accuracy of 45% in predicting enzyme classes. *Leijto et al. (2014)* based their work on *Borro et al. (2006)* and proposed an evolutionary system combining GA with Support Vector Machines (SVM) for the same purpose, achieving an accuracy of 71% and outperforming the previous classification methods. Although, the goal of POET is not protein classification, similar to the previous literature explored in this section, uses a novel evolutionary approach to evolve sequence function models.

*Wu et al. (2019)* proposed an ML-DE technique in which a combination of K-Nearest Neighbor, Linear Regression (LR), Decision Trees, Random Forests and Multi-Layer Perceptron (MLP) (from scikit library *Pedregosa et al., 2011*) approaches are used to generate sequence-function models able to predict how fit a protein is concerning a specified task. The best model is applied to evaluate proteins during an exhaustive computational mutagenesis. They evaluated their proposed method by finding fitter human GB1 binding proteins and show improvements over conventional DE methods. *Linder et al. (2020)* focused on improving the lack of diversity during mutagenesis for DNA and protein synthesis while not sacrificing fitness. They proposed a differentiable generative network architecture in which deep exploration networks are incorporated to generate diverse sequences by punishing similar sequence motifs. They employed a variational auto-encoder to ensure the generated diversity does not result in loss of fitness. They trained their system to design fitter proteins with regards to polyadenylation, splicing, transcription, and Green Flourescent Protein (GFP) fluorescence. In a performance evaluation on the same testbed, their proposed architecture designed fit sequences in 140 minutes while the classical approach of SA would need 100 days to achieve the same results. In 2021, *Repecka et al. (2021)* introduced ProteinGAN, a tool that uses Generative Adversarial Networks to learn important regulatory rules from the semantically-rich amino acid sequence space and predicts diverse and fit new proteins. ProteinGAN was experimented with to design highly catalytic enzymes, and the results show that 24% of their predicted enzyme variants are soluble and are highly catalytic even after going through more than 100 mutations. *Hawkins-Hooker et al. (2021)* emphasized how the availability of data regarding protein properties could be helpful for computational algorithms and trained their variational auto-encoders on a dataset consisting of approximately 70000 enzymes similar to luxA bacterial luciferase. They used Multiple Sequence Alignment (MSA) and raw sequence input for their system and showed that MSA better predicts distances in the 3D protein shape. They also diverged into predicting 30 new variants not originally in their dataset. *Cao et al. (2019)* and *Samaga, Raghunathan & Priyakumar (2021)* applied neural networks to predict the stability of proteins upon mutations. In 2021, *Das et al. (2021)* utilized deep generative encoders and deep learning classifiers to predict antimicrobials through simulating molecular dynamics. POET uses CEST contrast values of proteins as the function for evolving sequence-function models. The authors of this work could not identify a research which focuses in discovering such proteins at the time this document was prepared.

Evolutionary algorithms have been employed in motif extraction, function prediction, drug discovery, and directed evolution for a long time. *Yokobayashi et al. (1996)* applied a GA to replace mutagenesis and screening in DE. They used a dataset of 24 peptides with six amino acid sequences and their respective inhibitory activity levels. In their GA, each individual shared the same sequence as a data point of the dataset. The population of individuals went through recombination, while the fitness evaluation happened in wet labs with protein synthesis. They showed a 36% increase on average inhibitory levels of the population after six generations of their experiment. *Archetti et al. (2007)* incorporated four variants of GP (Tree-GP, Tree-GP with Linear Scaling for fitness evaluation, Tree-GP with constant input values and Tree-GP with dynamic fitness evaluation) to predict oral bioavailability, median oral lethal dose and plasma-protein binding levels of drugs of the dataset available in *Yoshida & Topliss (2000)*. They used Root Mean Square Error (RMSE) (*Willmott & Matsuura, 2005*) as their prediction error evaluation metric and compared their results with classic ML techniques (LR, Least Square Regression, SVM, and MLP), showing better performance on the GP side. *Seehuus, Tveit & Edsberg (2005)* proposed ListGP, a linear-genome Genetic Programming to discover important motifs from protein sequences. They employed the PROSITE dataset introduced in *Hulo (2004)* which represents motifs as regular expressions and contains information about the relevance between motifs and protein domains. ListGP showed improvements compared to Koza-style GP with Automatically Defined Functions for the task of classifying 69 protein families at 99% confidence interval level. Other evolutionary algorithms such as Immune GA (*Luo & Wang, 2010*) and the Multi-Objective Genetic Algorithm (*Kaya, 2007*) have also been applied to the problem of motif discovery before. *Chang et al. (2004)* utilized a Modified Particle Swarm Optimization (PSO) for discovering motifs in protein sequences. They translate amino acid symbols into numbers using a one-to-one translation table. Their results for two protein families of EGF and C2H2 Zinc Finger showed 96.9% and 99.5% accuracy, respectively. Much like the researches described in this paragraph, POET uses an evolutionary technique to evolve and produce sequence-function models able to predict the phenotypic performance of a given protein sequence. To do so, POET explores possible motifs found in the given input protein sequence and weighs them to determine the importance of these motifs while having a specific protein function in mind. Similar to *Yokobayashi et al. (1996)* POET models are utilized to replace parts of the Directed Evolution to accelerate the process. A key difference between POET and other works done in the field is incorporating a cycle made of both computational and wet-lab experiments. The sequence-function models evolved by POET are used to predict new potential candidates (with respect to a specified function) and these candidates are synthesized and experimented to validate the predictions and also to enhance the initial dataset of the proteins by adding more data points.

Availability of the relevant data is critical for training or evolving protein structure predicting models. Uniprot (*Uniprot Consortium, 2018*) is a massive dataset of protein amino acid sequences and their respective names, functions, and various structural properties containing more than 500,000 reviewed and more than 200,000,000 unreviewed entries. Proteomes[2] in this dataset belong to Bacteria, Viruses, Archaea and Eukaryota.

[2] A complete set of expressed proteins by an organism.

Brenda (*Chang et al., 2020*) is a dataset containing functional enzyme and metabolism data. This dataset consists of more than five million data points for approximately 90,000 enzymes in 13,000 organisms. Structural information of proteins, their metabolic pathways, enzyme structures, and enzyme classifications are among the data found in this dataset. Unlike most previous efforts, POET does not use a publicly available dataset. In fact, the initial dataset used for this experiment only consisted of 42 data points and one of the contributions of this research is to provide a comprehensive dataset of protein sequences and their respective CEST contrast values.

While researches in the field of motif discovery and ML-DE are closer to what POET does, other methods and algorithms are also discussed above to show promising results for applying computational approaches in protein engineering. These results aid protein engineers and computer scientists by discovering unknown protein properties and showing how the challenges in the field could be computationally formulated. Furthermore, some contributions evaluate the performance of different computational algorithms on the same general problem. Computer-aided protein engineering and optimization reduce the cost and time constraints of this line of research and enable exploring parts of the protein search landscape not easily achievable before. In this article, we propose the Protein Optimization Engineering Tool (POET), a Genetic Programming tool to aid protein engineers with finding potent protein variants concerning a specified function. Specifically, POET can be compared to algorithms used for Machine Learning-guided Directed Evolution. In the following subsection, we discuss Directed Evolution and ML-DE in detail.

# MATERIALS AND METHODS

We employ Genetic Programming as the computational problem solver of POET. In the following sections, different parts of POET are explained in more detail. Complete details on the wet-lab experiments and the methods used for synthesizing and evaluating the proteins can be found in (*Bricco et al., 2022*).

## Representation and Models

The Genetic Programming representation used in POET is a table of rules with four columns (Fig. 3). Each rule consists of a unique rule ID, a motif, a weight, and a status bit denoting whether the associated motif has been previously found in protein sequences of the dataset or not. It is essential to track the status of rules since only rules with the status of "1" are expressed for model evaluation or sequence prediction. Unexpressed rules with the status of "0" might be altered by undergoing recombinational operators in later generations and become useful once their motifs become less random and are found in the training dataset. Protein Optimization Engineering Tool models are initialized with constrained random motifs and weights in the first generation of evolution.

Evolved POET models predict the CEST contrast levels of a given protein sequence in the manner described in Algorithm 1. First, the input protein sequence is searched for motifs available in the model's rule table. Once the search is completed, the sum of the weights of the found motifs represents the CEST contrast prediction made by the predicting model. POET always prioritizes longer amino acid motifs over shorter ones. For

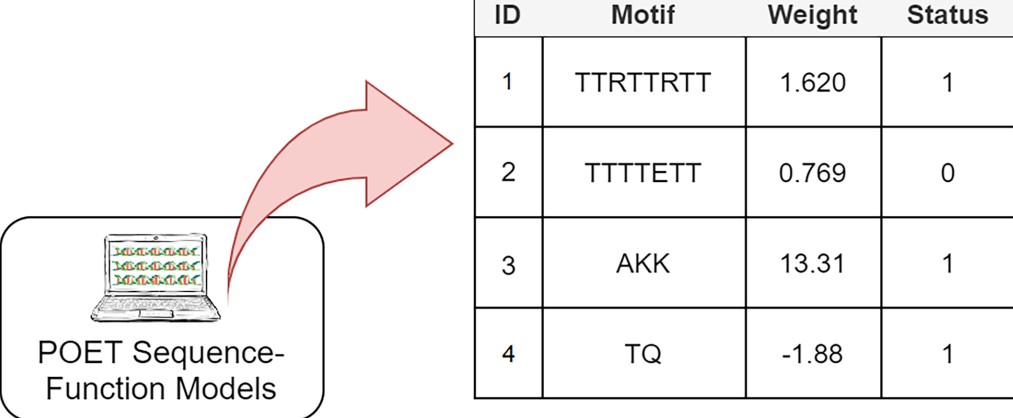

| ID | Motif | Weight | Status |
|----|-------|--------|--------|
| 1 | TTRTTRTT | 1.620 | 1 |
| 2 | TTTTETT | 0.769 | 0 |
| 3 | AKK | 13.31 | 1 |
| 4 | TQ | -1.88 | 1 |

POET Sequence-
Function Models

**Figure 3   A simple POET model consisting of four example rules with sequences and values obtained from an early POET model.** Each symbol in the motif sequence represents one amino acid. Rules 1, 3, and 4 have a status of 1 and are found in dataset. Rule 2 is not found in the dataset and is not expressed for evaluation and prediction purposes. Weights can be negative or positive float values.

example, if the protein sequence in hand is "TKW" and all "T," "K," and "TK" motifs are present in a model table, "TK" is prioritized over "T" and "K" rules and the position index points to "W" ignoring both of the shorter rules. As shown in the pseudo-code, the reverse sequence of the same motif is also evaluated for each rule. This is an attempt to extract more meaningful information from the abstract sequence space while exploring more of the search space in one go. Imagine a model which has a rule with the motif of "AKQY" which would have a reverse sequence of "YQKA". Considering the reverse sequence of this rule will remove the need for having another rule with the motif of "YQKA" in the model, leaving more space in the model's rule table for new motifs to be found during evolution. Furthermore, it is computationally less expensive to store and use both motifs as the same rule. Also in CEST contrast, this assumption is reasonable since the interactions that lead to contrast should only involve the functional groups and the peptides that we worked with are too small to allow for the formation of complex secondary and tertiary structures.

**Data**: *sequence, model*
**Result**: *predictedCEST*
*predictedCEST* ← 0 ;
*position* ← 0 ;
**while** *position* < *length(sequence)* ;      /* Loop through sequence symbols */
**do**
    **for** *rule: model.rules;*            /* Loop through model rules */
    **do**
        **if** *status is 0;*                /* Unexpressed rule */
        **then**
          | *continue*;
        **end**
        **if** *length(rule.pattern) + position* > *length(sequence)* **then**
         | *continue*;             /* sequence is not long enough */
        **end**
        *motif* ← *rule.motif*;
        *reversedMotif* ← *reverse(motif)* ;
        *portion* ← *sequence[position : (position + length)]* ;
        **if** *motif* = *portion* or *reversedMotif* = *portion;*    /* motif is found */
        **then**
          | *predictedCEST* ← *rule.weight + predictedCEST* ;
          | *break* ;
        **end**
    **end**
    *position* ← *position* + 1;
**end**
*return predictedCEST* ;

**Algorithm 1:** Pseudo-code for computing the predicted CEST contrast levels using a POET model. length() represents a function returning the size of a given input string array. This algorithm takes a protein sequence and a predicting model as input to output the predicted CEST contrast value.

## Selection mechanism and evolutionary operators used in the GP

Tournament selection (*Miller, Brad & Goldberg, 1995*) with elitism is used as the selection mechanism of POET. The individual with the highest fitness will always be selected for the next generation with no change (elitism). The rest of the population undergoes a tournament selection in which a pool of five individuals is randomly chosen, and the two fitter individuals are selected for performing crossover and creating a new offspring for the next generation.

**Crossover:** In the POET's crossover mechanism, every two fit parents chosen by tournament selection are used to generate one new offspring. All expressed rules (rules with a status bit of 1) and 20% of unexpressed rules for both parents are selected to form the new offspring. In order to avoid bloat, if the total number of rules in the offspring model

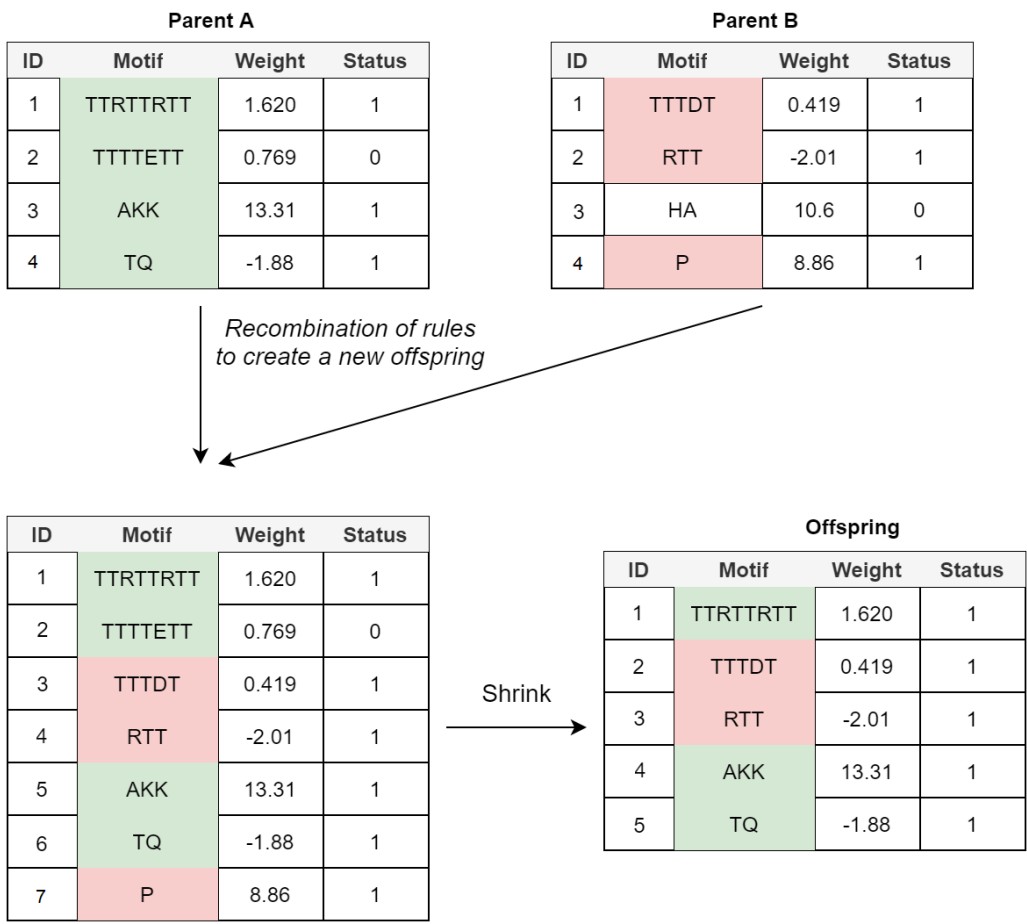

**Figure 4  A simple crossover example in POET where the maximum allowed rule count for each model is 5.** All the parents' expressed rules are selected to form the offspring. Unexpressed rules have a 20% chance of being selected while undergoing recombination. In this example, only unexpressed rule 2 of parent A is selected while the unexpressed rule 3 of parent B is not. Recombination of rules forms a new offspring table with size 7. Since the offspring size exceeds the allowed rule count by 2, two rules are removed during the model shrinking process. Unexpressed rules are prioritized to be removed in such a case. Since after removing all the unexpressed rules, the offspring still does not have a legal size, the shortest expressed rule is removed to form the final offspring.

table exceeds the maximum allowed size (an arbitrary value representing how large POET models can get with respect to number of rules), POET uses a shrink step to cut down the number of unexpressed rules and, if necessary, some of the expressed rules (prioritizing removing shorter rules over more extended rules) for the model to reach the permitted model size. At every step, model tables are sorted, arranging from longer motifs to shorter ones to follow the same design principle of prioritizing the discovery of longer motifs. Figure 4 illustrates a simple example of crossover in POET.

**Mutational Operators:** Multiple mutational operators are used in POET, all of which have a chance to increase the diversity of the models and help explore different valleys of the fitness landscape:

1. *Add Rule Mutation (ARM):* Adds a randomly generated rule to the model.
2. *Remove Rule Mutation (RRM):* Randomly selects a rule and removes it from the model.
3. *Change Weight Mutation (CWM):* Alters the weight of a randomly selected rule of the model by adding or subtracting a small random value (between 0 and 1) to/from it (equal chance).
4. *Add to Pattern Mutation (APM):* Randomly selects a rule and adds a amino acid symbol to its pattern.
5. *Remove from Pattern Mutation (RPM):* Randomly selects a rule and removes a symbol from its motif.

POET uses Algorithm 2 to apply these mutational operators on individual models.

**Data**: *population*

**Result**: *population* ;                                          /\* Mutated population \*/

**for** *individual in population* ;                     /\* Loop through population \*/

**do**

    **if** *rand*(0, 1) <*ARM rate* ;                     /\* Add Rule Mutation \*/

    **then**

        | *individual* ← *ARM*(*individual*);

    **end**

    **if** *rand*(0, 1) <*RRM rate* ;                     /\* Remove Rule Mutation \*/

    **then**

        | *individual* ← *RRM*(*individual*);

    **end**

    **for** *rule in individual.rules* ;              /\* Loop through individual rules \*/

    **do**

        **if** *rand*(0, 1) <*CWM rate* ;              /\* Change Weight Mutation \*/

        **then**

            | *rule* ← *CWM*(*rule*);

        **end**

        **if** *rand*(0, 1) <*APM rate* ;              /\* Add to Pattern Mutation \*/

        **then**

            | *rule* ← *APM*(*rule*);

        **end**

        **if** *rand*(0, 1) <*RPM rate* ;              /\* Remove from Pattern Mutation \*/

        **then**

            | *rule* ← *RPM*(*rule*);

        **end**

    **end**

**end**

**Algorithm 2:** Pseudo-code of the mutation evolutionary operator of POET. ARM and RRM happen on individual models, while CWM, APM, and RPM can mutate every rule of an individual. Each of these mutational operators has a configurable mutation rate.

## Evaluation of models

RMSE is used as the metric to evaluate the fitness of the POET models. During model evaluation, RMSE is calculated by measuring the error between the predicted and the actual values for the CEST contrast of all the proteins in the dataset. RMSE is an error measurement, and therefore lower values of it indicate accurate predictions. Since error values are squared when using RMSE as the fitness metric, models with high error rates are punished more than those whose error rate is lower. The following equation describes the RMSE formula:

$$RMSE = \sqrt{\frac{1}{n} \Sigma_{i=1}^{n} \left(d_i - f_i\right)^2}$$

where $n$ is the number of data points, $i$ is an index referring to each individual, $d_i$ is the measured value and $f_i$ is the predicted value for individual $i$. To better train POET models starting with a relatively small dataset, 10-fold cross-validation (*Arlot, Sylvain & Celisse, 2010*) was used. In k-fold cross-validation, the dataset is divided into k groups, and each model gets evaluated k times. Each time, a group is selected to act as the test set while the rest work as the training set. This process continues until all groups are selected as the test set. The individual's fitness is the average fitness among all the k iterations. K-fold cross-validation helps evolve more accurate and generalized models, especially if the dataset is small.

```
Data: model, symbols, generations
Result: population ;                          /* Predicted protein population */
population ← init_random() ;       /* Randomly initialize the population */
gen ← 0;
while gen < generations do
    for sequence in population do
        old_sequence ← sequence;
        random_site ← rand(0, length(sequence));
        random_symbol ← rand_choice(symbols);
        sequence[random_site] ← random_symbol ;      /* Alter a random site */
        if model.predict(sequence) < model.predict(old_sequence) then
            sequence ← old_sequence ;         /* Keep the altered sequence */
        end
    end
    gen ← gen + 1;
end
sort(population);
return(population);
```

**Algorithm 3:** Evolving protein sequences using a trained Protein Optimization Engineering Tool sequence-function model. rand_choice() chooses a random element from a given list of elements. model.predict() predicts the fitness of a given protein sequence using the best previously trained model. Starting from a random population of sequences increases the novelty in prediction.

## Optimization and prediction of peptides that produce high CEST contrast

The proposed system is a multi-epoch feedback system, here used to predict better proteins concerning their CEST contrast level measured by MRI. As illustrated in Fig. 2, in each epoch of the experiment, a dataset containing protein sequences and their respective CEST contrast values evaluated in MRI is given to POET. POET computationally evolves protein sequence-function models over generations, and the fittest model is selected for the next step. Then a pool of randomly generated protein sequences is given to the fittest POET model for evaluation. A copy of the best sequence regarding predicted fitness is saved without change (elitism), and the pool of the protein variants undergo mutagenesis, with every mutant being only one symbol away from their parent protein. If there is no improvement after mutation of a sequence, the old sequence reverts, and no changes are applied to that individual. This process repeats for an arbitrary number of generations. Afterward, the fittest predicted proteins in the sequence pool of the latest generation are chosen for wet-lab measurements. Finally, the new data points are added to the dataset to improve the POET models' accuracy and learn from previous epochs. Algorithm 3 exhibits the details of this implementation.

## External knowledge on soluble proteins

Medical characteristics of protein engineering make it very important for CEST protein agents to be soluble in water. Therefore, applying a simple hydrophobicity threshold improves finding soluble proteins and better predictions. We use the data shown in Table 1. If the sum of the hydrophobicity levels of amino acids in a peptide sequence is less than zero, we consider the peptide insoluble and do not use that sequence in protein optimization and prediction (sequence fitness is set to zero). This step is applied during the prediction of new proteins in which a population of protein sequences are evolved to discover proteins that potentially have high CEST contrast values. In the prediction step, if a generated sequence does not follow the hydrophobicity rule (sum of hydrophobicity levels is below zero), it will be considered to have a CEST contrast of zero and is not evaluated using an evolved model.

## Dataset and code availability

For the first epoch of the experiment, a dataset containing only 42 data points derived from available literature was used. After that, at least ten new data points were added in each epoch dataset through wet-lab experiments based on the predicted proteins. In the final epoch of the experiment, the dataset contained 159 data points of protein sequences and their respective CEST contrast values, with some of the new variants being wild types. Table 2 shows the dataset curated during POET epochs. The curated dataset as well as the source code for running POET experiments written in the Python programming languages are provided with this article.

**Table 1 Table of amino acids and their respective hydrophobicity values (*Rose et al., 1985*).**

| Symbol | Hydrophobicity | Symbol | Hydrophobicity |
|--------|----------------|--------|----------------|
| I | −0.31 | Q | 0.58 |
| L | −0.56 | C | −0.24 |
| F | −1.13 | Y | −0.94 |
| V | 0.07 | A | 0.17 |
| M | −0.23 | S | 0.13 |
| P | 0.45 | N | 0.42 |
| W | −1.85 | D | 1.23 |
| J | 0.17 | R | 0.81 |
| T | 0.14 | G | 0.01 |
| E | 2.02 | H | 0.96 |
| K | 0.99 | | |

## RESULTS

### Experimental setup

The experiments were performed for eight epochs. Multiple experiments were run with different configuration of parameters and the ones that produced the best results were chosen. Table 3 shows a brief overview of the used experimental parameters of POET for the case study. Experiments were conducted on Michigan State University's High-Performance Computing Center (HPCC) computers details of which are available in (*ICER, 2022*). Each of the experiment repeats were run in parallel using a single HPCC CPU core (2.5 GHz) and 8 GB of RAM on a single node.

### Evolution of models per epoch

In each epoch of the experiment, POET is run 50 times, each repeat evolving a population of 100 models over 10,000 generations. The overall best model is used to predict the protein sequences to be tested in a wet lab at the end of each epoch. Table 4 summarizes the performance of the trained models throughout the experiments. *Training RMSE* is the RMSE calculated for each epoch dataset. For example, to predict the proteins of epoch 7, models were trained using all the available data of epochs 1 to 6. *Test RMSE* is calculated for the training set coming from the 10-fold cross-validation. *Best Overall RMSE* is measured for the best models of each epoch on all available data and not only the epoch data. Finally, *Average Overall RMSE* indicates the RMSE levels of the 50 best models of each epoch on average against all available data. Test RMSE for all the epochs is slightly greater than the training fitness for the same epoch showing a slight over-fitting of the models. Each POET model can have a maximum number of 100 rules that can be either expressed or unexpressed. There were no assumption on the number of expressed or unexpressed rules for each model, however, it is interesting to report these values since they could be a ground for further future analysis. The best model of each Epoch approximately uses all of the possible 100 rule space while almost 50% of those rules are expressed (on average around 44 rules out of 97 are expressed). Experiments performed for each epoch are not built upon the previous evolved models. In other words, the evolution of models start with a

Miralavy et al. (2022), *PeerJ Physical Chemistry*, DOI 10.7717/peerj-pchem.24

**Table 2 All the available data in the dataset used for all the epochs of POET including the mock test data and the discovered protein sequences.** All the data points with epoch $N$ and less than $N$ were used to evolve models and predict the data points marked as $N + 1$.

| # | Sequence | CEST contrast value (3.6 ppm) | Epoch | # | Sequence | CEST contrast value (3.6 ppm) | Epoch | # | Sequence | CEST contrast value (3.6 ppm) | Epoch | # | Sequence | CEST contrast value (3.6 ppm) | Epoch |
|---|---|---|---|---|---|---|---|---|---|---|---|---|---|---|---|
| 1 | KKKKKKKKKKKK | 12.5 | 1 | 45 | GIFKTTKCKHNS | 7.61 | 2 | 89 | HDDKNKESDD | 7.479407 | 6 | 133 | KPCKWAGRACAK | 16.69529 | TEST |
| 2 | KSKSKSKSKSKS | 17 | 1 | 46 | SNHKMSECRGLR | 5.98 | 2 | 90 | QERRDDILWD | 2.296864 | 6 | 134 | CQLAWRPCAKAS | 19.532 | TEST |
| 3 | KHKHKHKHKHKH | 12.7 | 1 | 47 | FNSNKITPTSNM | 5.29 | 2 | 91 | KRIIEDDQLE | 15.19034 | 6 | 135 | QCAGWVQKRQIQ | 23.37685 | TEST |
| 4 | KGKGKGKGKGKG | 10.8 | 1 | 48 | VNSDPSNGQMRD | 4.15 | 2 | 92 | VCNRIEPLKPIL | 21.82239 | 7 | 136 | RRCQAQEFWLGA | 9.515134 | TEST |
| 5 | KSSKSSKSSKSS | 13.2 | 1 | 49 | LSNRRGREQYAG | 7.08 | 2 | 93 | LHSSQWLKVDHLL | 18.17873 | 7 | 137 | GLIEARAMQQCC | 2.704976 | TEST |
| 6 | KGGKGGKGGKGG | 11.8 | 1 | 50 | QTATENSQMNSG | 3.64 | 2 | 94 | VINKVISNPCVN | 8.107188 | 7 | 138 | QCRAGAMPAMYV | 12.05809 | TEST |
| 7 | KSSSKSSSKSSS | 13 | 1 | 51 | QTEHYENSARNS | 1.09 | 2 | 95 | GNKKNWRWYKNR | 14.71334 | 7 | 139 | NFLRAQRQCQKQ | 18.16368 | TEST |
| 8 | KGGGKGGGKGGG | 12.1 | 1 | 52 | KDRTSKPKRPWC | 8.67 | 3 | 96 | ICLKSQPICGID | 29.49547 | 7 | 140 | AQCCQHRKGYMN | 14.69376 | TEST |
| 9 | RRRRRRRRRRRR | 22 | 1 | 53 | GRKRGAIWKDTK | 12.75 | 3 | 97 | LWSDIKMKLKKT | 49.37196 | 7 | 141 | NRVTESVRNVKM | 3.683273 | TEST |
| 10 | RSRSRSRSRSRS | 12.8 | 1 | 54 | CCWHNPKWRRTR | 18.46 | 3 | 98 | NWRDCLSLIVPN | 3.179373 | 7 | 142 | NVVVQRRNHHTS | 28.05341 | TEST |
| 11 | RGRGRGRGRGRG | 17.2 | 1 | 55 | KYTKTRKQSSKA | 22.48 | 3 | 99 | KMGKLIGIPVLK | 47.83688 | 7 | 143 | VINKVISCPCVN | 8.109024 | TEST |
| 12 | RHRHRHRHRHRH | 5.5 | 1 | 56 | RGKMPLRWMTRK | 17.14 | 3 | 100 | NDISMCNKNNNW | 8.824446 | 7 | 144 | GGRVWEWNVAA | 6.08351 | TEST |
| 13 | RTRTRTRTRTRT | 18.7 | 1 | 57 | GNCPMKVCSPMG | 8.89 | 3 | 101 | VSLQCWELGPNK | 15.70919 | 7 | 145 | NNKCQVVAAFVM | 5.401218 | TEST |
| 14 | RTTRTTRTTRTT | 16.3 | 1 | 58 | VNLPMVMPNLRM | 4.53 | 3 | 102 | TVSEPVMMVSVS | 7.771304 | 8 | 146 | VLTWSAVNNNVQ | 0 | TEST |
| 15 | RTTTRTTTRTTT | 18.9 | 1 | 59 | GPMPMNAKMKLC | 5.81 | 3 | 103 | PRSWEVKEKETM | 18.27337 | 8 | 147 | NCGVNLVNAVGQ | 0 | TEST |
| 16 | TTTTTTTTTTTT | 6.5 | 1 | 60 | KVIRYVVAPMKL | 8.94 | 3 | 104 | PGGVRSNDLLEV | 11.18031 | 8 | 148 | HIAVVNWVNVGH | 0 | TEST |
| 17 | TKTKTKTKTKTK | 14.1 | 1 | 61 | IKGMNIKMPTDQ | 9.95 | 3 | 105 | PVNRLGKMSKNR | 28.83256 | 8 | 149 | CNNIQGRNNSVW | 0 | TEST |
| 18 | DTDTDTDTDTDT | 2.2 | 1 | 62 | MWQMKWTRKTRE | 16.24 | 4 | 106 | VGSVKSGNLRMR | 26.22986 | 8 | 150 | VPNIQVKGSK | 4.99326 | TEST |
| 19 | ETETETETETET | 1.7 | 1 | 63 | HGRKWKRTKFDD | 15.49 | 4 | 107 | TSKSKKRMTAKK | 29.83349 | 8 | 151 | PVARKVVQICHP | 18.63698 | TEST |
| 20 | TTKTTKTTKTTK | 12.6 | 1 | 64 | DKRKIKQKMWWG | 10.86 | 4 | 108 | ETNVRVKVVSES | 5.707696 | 8 | 152 | VTRMTIQVKGSK | 30.16713 | TEST |
| 21 | DTTDTTDTTDTT | 4 | 1 | 65 | RRMVNRTITRMW | 15.01 | 4 | 109 | EPSNLPKGMNEK | 24.69024 | 8 | 153 | MAMADAAAPMNA | 6.965346 | TEST |
| 22 | ETTETTETTETT | 4.4 | 1 | 66 | HWSTCTRTRTLS | 17.1 | 4 | 110 | RLWNSGEGRGEN | 12.26758 | 8 | 154 | MKVAAAMAPKQV | 38.66109 | TEST |
| 23 | TTTKTTTKTTTK | 13.8 | 1 | 67 | WWWKPKREDFMK | 6.58 | 4 | 111 | ELNTGLVLVNWK | 0 | 8 | 155 | PVVYKTVIQCCD | 4.333159 | TEST |
| 24 | DTTTDTTTDTTT | 4 | 1 | 68 | HIKWRLTKGTRT | 16.08 | 4 | 112 | RPPMLNVVRVVG | 6.489129 | TEST | 156 | KVLWRMPAQIIQ | 13.51793 | TEST |
| 25 | ETTTETTTETTT | 4 | 1 | 69 | WDRTSTRPSSVL | 13.46 | 4 | 113 | KWVVRPRIRRLL | 14.62156 | TEST | 157 | VSVVATGCVWET | 14.33556 | TEST |
| 26 | TTTTTKTTTTTK | 13.8 | 1 | 70 | KPWHGCASRTKR | 16.19014 | 4 | 114 | IGVLRSVKQTVR | 28.55074 | TEST | 158 | AKCKVQSANVCK | 37.82153 | TEST |
| 27 | DTTTTTDTTTTT | 7.2 | 1 | 71 | KKRLHWIRWHCG | 12.01536 | 5 | 115 | VINKVISNPCVN | 8.516833 | TEST | 159 | VAWVMKAHVCTM | 4.201038 | TEST |
| 28 | ETTTTTETTTTT | 5.9 | 1 | 72 | RKHHGWRWEQWK | 13.59362 | 5 | 116 | ETNVRVKVVSES | 2.368102 | TEST | | | | |

Miralavy et al. (2022), *PeerJ Physical Chemistry*, DOI 10.7717/peerj-pchem.24

**Table 2** (*continued*)

| # | Sequence | CEST contrast value (3.6 ppm) | Epoch | # | Sequence | CEST contrast value (3.6 ppm) | Epoch | # | Sequence | CEST contrast value (3.6 ppm) | Epoch | # | Sequence | CEST contrast value (3.6 ppm) | Epoch |
|---|---|---|---|---|---|---|---|---|---|---|---|---|---|---|---|
| 29 | DSDSDSDSDSDS | 2.5 | 1 | 73 | WFGLQRHLKKKD | 19.0715 | 5 | 117 | RLPKRVQGNVEK | 30.61573 | TEST | | | | |
| 30 | DSSSDSSSDSSS | 7 | 1 | 74 | CHLKDLRKMGLR | 10.1388 | 5 | 118 | GLGNQHVVVLGV | 3.528919 | TEST | | | | |
| 31 | DSSSSSDSSSSS | 9.1 | 1 | 75 | KMWDWEQKKKWI | 34.11149 | 5 | 119 | KVRCLVEARPSW | 8.194995 | TEST | | | | |
| 32 | KKRKKHKKGKKP | 11.9 | 1 | 76 | QRHDSHRHGLWL | 7.543669 | 5 | 120 | HLVVSPRVSWGC | 5.30282 | TEST | | | | |
| 33 | KKAKKKGKKHKK | 9.9 | 1 | 77 | LELKLGKRPMGW | 29.24477 | 5 | 121 | IIRSPICCVSRV | 12.89188 | TEST | | | | |
| 34 | KKGKKKGKKHKK | 11.3 | 1 | 78 | GQRWLYKMKDSM | 11.86265 | 5 | 122 | DKRKIKQKMWWG | 19.13841 | TEST | | | | |
| 35 | KKGKKKGKKPKK | 9.3 | 1 | 79 | MWVKGMKHKKMK | 13.23495 | 5 | 123 | RKHHGWRWEQWK | 20.00466 | TEST | | | | |
| 36 | MPRRRSSSRPVRRRR PRVSRRRRRRGGRRRR | 19 | 1 | 80 | LDHTWGKWGHQS | 11.50256 | 5 | 124 | EMRQWKWMWENA | 6.580628 | TEST | | | | |
| 37 | DWNNYLYQNLH | 0 | 1 | 81 | DKVCKIQKRKWH | 12.51172 | 5 | 125 | PIKQIAWPIIEH | 13.6599 | TEST | | | | |
| 38 | SYYWLWWHQQI | 0 | 1 | 82 | WDWEQKKKWI | 31.26163 | 6 | 126 | KMWDWEQKKKWI | 24.09143 | TEST | | | | |
| 39 | NWNWWGLSYLA | 0 | 1 | 83 | ERQEEKIKKW | 20.79427 | 6 | 127 | ARNRKKIMMRWI | 29.01464 | TEST | | | | |
| 40 | NQYSNWNKNYK | 6.94 | 1 | 84 | SDGSKIKDRD | 8.630329 | 6 | 128 | NAPWKHWRIINE | 8.898582 | TEST | | | | |
| 41 | NENQWHYYWRQ | 0 | 1 | 85 | SSDQDRDKWL | 16.82505 | 6 | 129 | NKQRRMLSRERS | 28.52089 | TEST | | | | |
| 42 | NGTLYLNNYYE | 0 | 1 | 86 | LLRLLGLVER | 3.037419 | 6 | 130 | LSQQPRKRATWR | 12.60838 | TEST | | | | |
| 43 | NSSNHSNNMPCQ | 14.06 | 2 | 87 | KEEVWLKWLI | 13.42497 | 6 | 131 | IRRWNDRIRITS | 13.90809 | TEST | | | | |
| 44 | IRTYLRKRNSTQ | 8.03 | 2 | 88 | KGKLDKDRNL | 23.28052 | 6 | 132 | MAALLYQHRLARR | 3.528221 | TEST | | | | |

**Table 3   Experimental parameters used for the case study.**

| Parameter | Value |
|---|---|
| Population size | 100 |
| Tournament size | 5 |
| Max rule motif length | 9 |
| Max table rule count | 100 |
| Number of generations | 10,000 |
| Unused rule crossover rate | 20% |
| Mutation rate | 16% for all types |

**Table 4   Comparison of performance of the overall best POET models at generation 10,000 for all the epochs.**

| Epoch | Data points | Training RMSE | Test RMSE | Best overall RMSE | Average overall RMSE | Total rules # | Expressed rules # |
|---|---|---|---|---|---|---|---|
| E1 | 42 | 1.272 | 1.307 | 10.189 | 16.583 | 97 | 44 |
| E2 | 51 | 1.558 | 1.576 | 11.001 | 15.882 | 97 | 45 |
| E3 | 61 | 2.238 | 2.258 | 10.185 | 13.276 | 96 | 45 |
| E4 | 71 | 2.308 | 2.326 | 10.096 | 12.786 | 96 | 44 |
| E5 | 82 | 2.898 | 2.919 | 8.974 | 11.531 | 97 | 44 |
| E6 | 92 | 3.651 | 3.674 | 8.247 | 11.349 | 96 | 44 |
| E7 | 102 | 3.923 | 3.944 | 8.686 | 10.646 | 97 | 44 |
| E8 | 112 | 4.891 | 4.916 | 7.845 | 9.954 | 97 | 44 |

random population from scratch in each epoch but with a larger dataset. An increase in the RMSE level of the best model is evident as the epoch number increases (Training RMSE column) showing the difficulty of finding fitter models on more data points. This increase does not indicate that models of later epochs are less accurate since the dataset used for evaluating each model is different. To support this claim and to show that the RMSE levels actually improve over epochs, it is interesting to see how well the same best models for each epoch perform when tested against all available data. When tested against all available data (Average Overall RMSE column), as the epoch number increases, the RMSE values for average of the models decreases, showing that the trained models are improving on average in each epoch (also illustrated in Fig. 6). As for the best models of each epoch, RMSE levels slightly increase for epoch 2 and epoch 7 compared to epoch 1 and epoch 6, respectively. However, RMSE of the best model in epoch 8 is lower than all previous epochs.

To further clarify the results, Fig. 5 illustrates the evolution of models for all eight epochs. The drop in RMSE values in early generations is visible in all the experiments and is due to starting with randomly initialized populations. The same evolutionary method is used throughout all the epochs; however, the number of data points in each epoch varies from the previous one due to the addition of new data at the end of each epoch. For example, for epoch 1, the training dataset had only 42 data points making it easier for the algorithm to achieve lower RMSE values by quickly over-fitting the data. Meanwhile, 111

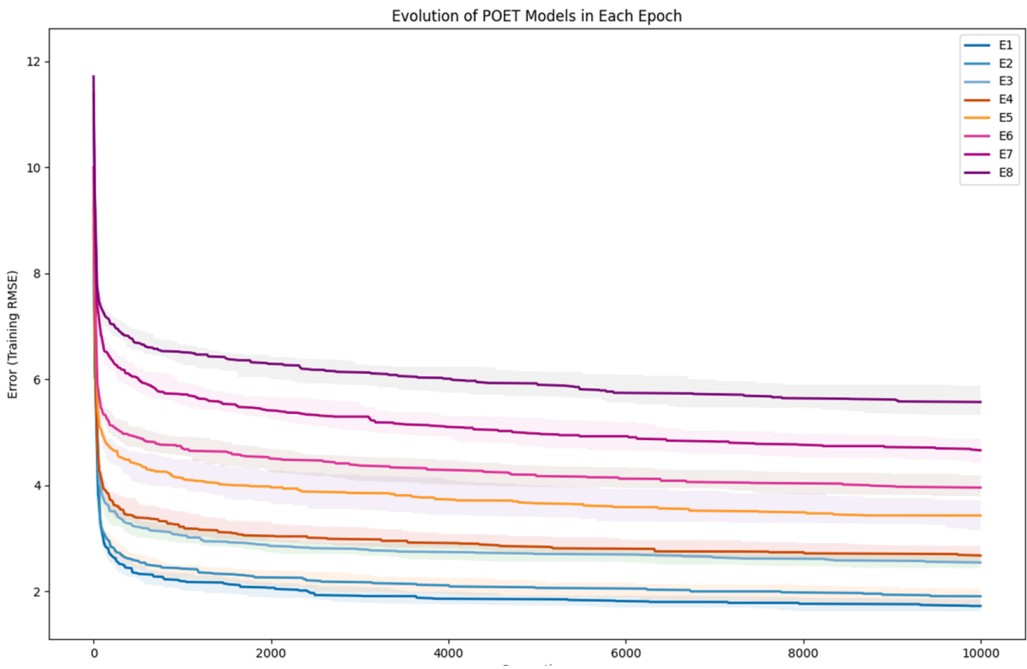

**Figure 5** **Evolution of the POET models over generations for eight epochs of the POET experiment.** The fitness metric is Training RMSE (RMSE over epoch data), and therefore, lower values indicate more accurate measurements.

data points were available in the dataset used for epoch 8, which explains higher RMSE values. Furthermore, in earlier epochs, the 50 best models performed more similarly, while models performance varies more in the later epochs.

In each epoch, after model training, the best POET model is used to choose fitter proteins in a large pool of artificially generated protein sequences. In this process, all the generated sequences are evaluated using the model and the top 10 sequences are subsequently selected to be tested in wet labs. A mock test was designed to analyze this process on a small dataset of 43 protein sequences with actual measured CEST contrast values. None of the data points in this set were used in the training of the best POET model (best model of E8). These data points were given to the best POET model to evaluate and sort based on their predicted CEST contrast values. For a perfect model, the order of proteins after sorting would be the same as the order of these proteins sorted based on their actual CEST contrast values.

Figure 7 shows predicted order made by the best model of Epoch 8 in orange and their actual order in the dataset in blue. Although not significantly, the two series positively correlate with Pearson correlation coefficient value of 0.63. In each POET epoch, 10 new sequences are chosen to be evaluated in wet labs. An interesting observation is that five out of the top 10 fittest protein sequences are among the top 10 predictions of the best model.

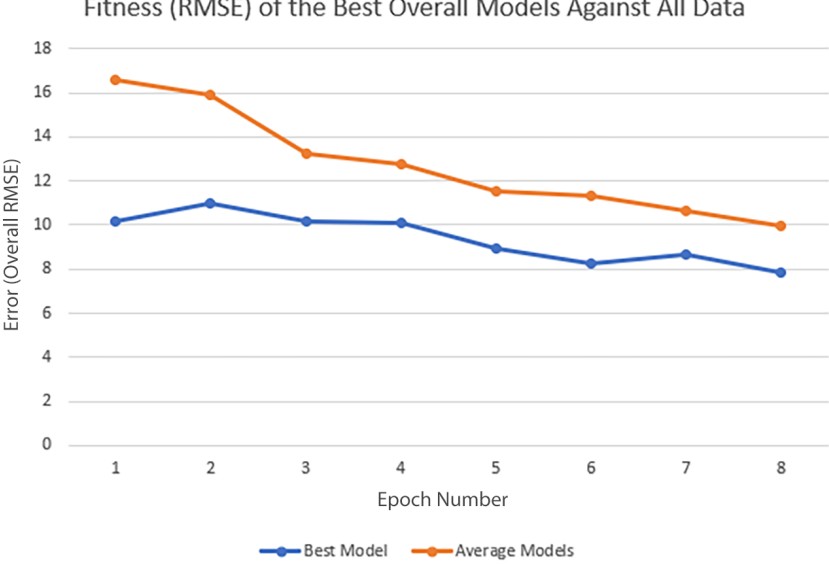

**Figure 6** **Improvement of the POET models in each epoch of the experiment.** Overall RMSE (RMSE over all available data) levels for the best models of each epoch (blue) and the average overall RMSE of all the models in each epoch (orange) are tested against all available data.

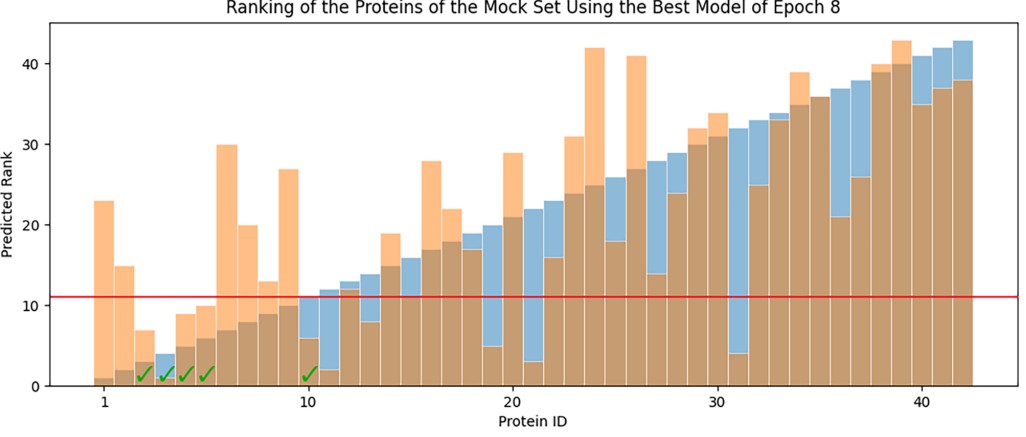

**Figure 7** **Predicted ranks (orange) of the protein sequences in a mock set *versus* their actual rank (blue).** The proteins under the red line were among the top 10. The check-marked bars indicate top 10 protein sequences which were also predicted by the best model as top 10.

## Improvement of CEST Contrast Proteins

While this article focuses on the computational aspects of this work, details of the wet-lab experiments and how these experiments are performed can be found in *Bricco et al. (2022)*. An essential aspect of the design goal of POET is to replace parts of DE in order to enhance the process by finding fitter proteins in a shorter time with less cost. We measure the CEST contrast levels during each epoch of our experiment. $MTR_{asym}$ [3] (*Wu et al., 2016*) is the most common metric for evaluating CEST contrast protein agents and determines the

[3] $MTR_{asym}$ refers to Asymmetrical Metric

signal strength of the target proteins in an MRI environment and is calculated through the following equation:

$$MTR_{asym}(+\tau) = \frac{S_{-\tau} - S_{+\tau}}{S_0}$$

in which $S_{+\tau}$ and $S_{-\tau}$ is the measured signal at spectral location $\tau$ with RF[4] saturation at two points of $+\tau$ and $-\tau$ respectively. $S_0$ is the exact measurement without RF saturation (*Wu et al., 2016*). We normalized MTR against K12, a peptide of 12 K (lysine) amino acids, to have a fair comparison across all cycles of our experiments. This peptide was chosen due to its high MRI contrast value and its similarity to the other reported results.

After the second cycle of our experiment, our predicted peptides, on average, had a higher contrast value than K12. On cycle seven, POET predicted a protein with an approximately 400% increase in contrast to K12 (Results available in *Bricco et al., 2022*).

Prior efforts to increase CEST contrast focused on increasing the charge, since positively charged amino acids, such as lysine, provide the exchangeable protons that produce contrast, and it was believed that the best way to increase contrast would be to increase the number of these exchangeable protons (*Farrar et al., 2015*). However, since POET's exploration is not limited to specific parts of the search space, we managed to find proteins that do not follow this principle and yet have a high CEST contrast value. Details of this experiment can be found in *Bricco et al. (2022)*.

Figure 8 demonstrates the most frequently observed motifs among the best models of the 50 repeats performed for epoch 8. 63 of the discovered motifs are common between at least 10% of the models in epoch 8. Motifs with a single amino acid symbol are the most trivial to find and therefore are the most common motifs among the models. In addition, four motifs with a length of higher than six symbols are found. All of these motifs are listed as the *x*-axis of Fig. 8.

## DISCUSSION AND FUTURE DIRECTIONS

The main contribution of this article is POET, a tool based on Genetic Programming to predict protein functions from the abstract and yet semantically rich representation of amino acid sequences of proteins and peptides. Directed Evolution starts mutagenesis from proteins with analogous properties to the desired one. Similar starting points make it highly likely for this process to get stuck in a local optimum. Unlike Directed Evolution, POET starts the prediction process from random sequences, enabling it to explore different areas of the protein search space possibly jumping between optima to evolve fitter models. Results indicate that it is possible to evolve fitter Chemical Exchange Saturation Transfer contrast predicting models over epochs, with the best model of Epoch 1 having an RMSE score of 10.189 over all available data, while the best model of Epoch 8 has an RMSE of 7.845 for the same dataset. Running a mock experiment to evaluate the best model of Epoch 8 (Best model selected for predicting sequences of the next epoch), this model was able to find five of the top 10 Chemical Exchange Saturation Transfer proteins. Comparing the ranks predicted by the best model and the actual ranks of the proteins in the dataset

[4]Radio-Frequency.

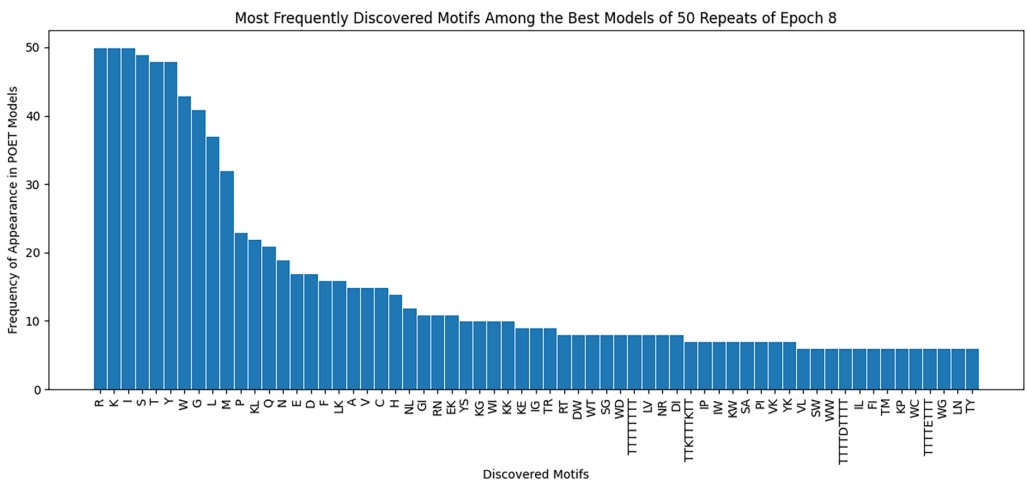

**Figure 8** **Frequency of discovered motifs in the best models of the 50 repeats of epoch 8.** Due to the simplicity and the high number of data points only motifs are shown that are common between at least 10% of the models.

with respect to their CEST contrast showed a positive correlation with Pearson $r$ value of 0.63.

Our findings demonstrate an improvement in the Chemical Exchange Saturation Transfer levels of the predicted proteins with the best value from prior work being 19 (salmon protamine), and POET discovering 27 peptides which produce higher contrast (see Table 2). During the experiments, a protein with four times higher CEST contrast than K-12 was discovered. Results on Fig. 8 suggests that there are common motifs discovered between the best-evolved models while some diversity persists. This diversity, along with the randomness of POET has been beneficial in our experiments by predicting a diverse range of protein sequences and guiding us to explore sections of the search space, which was not possible before using conventional Directed Evolution methods.

Another contribution of this work is curating a rich dataset consisting of 159 proteins and their respective Chemical Exchange Saturation Transfer contrast values (for 3.6 PPM) during the eight epochs of the experiment. As a side discovery, some of the predicted proteins indicated that fitter proteins do not necessarily follow conventional bio-engineering theories and by their relatively low concentration of positively charged amino acids. One notable example is NSSNHSNNMPCQ (*Bricco et al., 2022*), producing greater contrast than the existing reporter KKKKKKKKKKKK (*Gilad et al., 2007*), while having a neutral charge. This dataset is made available for researchers in this manuscript.

The same algorithm with the same number of generations under-performs in evolving accurate models as the dataset grows. The Protein Optimization Engineering Tool is a modular system with the potential of improvement on many fronts. For example, dynamic recombination rates and fitness evaluation by considering motif diversity in the population can be an excellent approach to improving models' evolution over generations. Incorporating ideas of active learning to enhance the prediction and selection of the

proteins to be tested in the wet labs could potentially improve the chances of finding fitter proteins in fewer epochs. Adding more structural information and proven natural rules to quickly explore more extensive parts of the search space is also a viable future direction. Although the focus of this research was on evolving CEST contrast proteins, the capabilities of POET is not limited to evolving sequence-function models of this type. Therefore, it will be interesting to test POET with datasets of different protein functionalities. Furthermore, each evolved POET model aims to fit a local optima. It would worthwhile to try an ensemble method which uses a combination of POET models to more accurately predict the CEST contrast values of given protein sequences.

## ACKNOWLEDGEMENTS

Computational model evaluation was done on MSU's HPCC system and is gratefully acknowledged.

### Funding
Support for this work came from NIH under award nrs. 1R01EB030565-01, 1R01EB031008-01 and P41-EB024495. There was no additional external funding received for this study. The funders had no role in study design, data collection and analysis, decision to publish, or preparation of the manuscript.

### Grant Disclosures
The following grant information was disclosed by the authors:
NIH: 1R01EB030565-01, 1R01EB031008-01, P41-EB024495.

### Competing Interests
Wolfgang Banzhaf is an editor for PeerJ.

### Author Contributions

- Iliya Miralavy conceived and designed the experiments, performed the experiments, analyzed the data, performed the computation work, prepared figures and/or tables, authored or reviewed drafts of the article, and approved the final draft.
- Alexander R. Bricco conceived and designed the experiments, performed the experiments, analyzed the data, authored or reviewed drafts of the article, and approved the final draft.
- Assaf A. Gilad conceived and designed the experiments, analyzed the data, authored or reviewed drafts of the article, and approved the final draft.
- Wolfgang Banzhaf conceived and designed the experiments, analyzed the data, authored or reviewed drafts of the article, and approved the final draft.

### Data Availability
The input sequence-function dataset and the code for replicating the results of our experiments are available in the Supplemental Files.

## Supplemental Information

Supplemental information for this article can be found online at http://dx.doi.org/10.7717/peerj-pchem.24#supplemental-information.

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
