# Peer review of "Using genetic programming to predict and optimize protein function"

_PeerJ Physical Chemistry, doi:10.7717/peerj-pchem.24_

## Round 0.1 · original submission · Major Revisions

Thank you for your patience with the review process. I had an extremely difficult time finding suitable reviewers for your manuscript. I am glad that we have now two useful reviews. Please address the concerns of the reviews in your revisions.

Reviewer 1 ·

Basic reporting

The article is comprehensible overall, but it could be improved by addressing the comments about content and formatting below.

Literature references and background/context:
Line 134 (Related Works section): While having a comprehensive list of references can be useful, most of them are described in a single sentence. As a result, only a general description of each reference is provided, and their relationship to the current work is often unclear.

This section would be more valuable if the relationships between the references and the research reported in the paper were explicitly described, with sufficient detail to illustrate how the authors' approach improves upon weaknesses of previous approaches.

For example, many of the cited algorithms are aimed at classifying proteins according to structure or predicting protein structure. Their relationship to and utility for protein engineering applications are not clear. On the other hand, algorithms for identifying motifs appear to be relevant, but this was not obvious to me until I read the methods section.

Discussing the strengths and weaknesses of references that are more closely related to the reported research would enable readers to see how the authors' POET algorithm improves upon previous work.

Presentation of Figures and Tables:
Both Fig. 5 and 6 plot RMSE versus Epoch number, but they use different types of RMSE’s (4 different RMSE’s are listed in Table 4). This is not clear from the plots or figure legends. The RMSE on the axes of each plot should be given different names.

Figure 1: explain that the image on the right is a fitness landscape, with orange dots representing variants in the initial library and pink dots representing improved variants. It would be a good idea to add a cluster of orange dots to the fitness landscape in panel A to represent the library. This would more clearly show that the search space in experimental DE is limited.

Fig. 5: showing the epoch scale going up to 9 is confusing, since only 8 epochs were run

Raw data availability:
Source code and final data set are included as supplemental files, but their availability is not mentioned in the manuscript. The downloaded Learn_Dataset.csv appears to have 154 entries, but the methods section (line 327) says the final data set should have 163 entries.

Minor clarifications in language:
The manuscript has a few typos (missing spaces (line 164 and a few other places), missing commas).

Line 161 (…”to predict the DEAD box family”): Do you mean predict membership in the family of DEAD box proteins?

Line 210: This is the first time the acronym RMSE is used, and it should be defined here.

Line 396 – Discussion section: There is some redundancy between the results and discussion. Some statements in this section repeat conclusions that were discussed in the results section (rather than discussing them in the discussion section). For example, line 419 states that some of the identified proteins that had improved CEST values “do not necessarily follow conventional bio-engineering theories”, but the discussion does not explain what the conventional theories are. There is some explanation in the results section on this topic (line 388), but why positively charged amino acids would improve CEST contrast was not explained in terms of a biophysical theory. Also, line 141 states that the “findings demonstrate an improvement in the CEST levels of the predicted proteins”, but this is not shown clearly in either the results or discussion sections (see below).

Experimental design

The authors describe the development and application of a genetic programming algorithm that performs in silico mutagenesis and screening to identify protein variants that score well against a model describing sequence motifs associated with a function (in this case, motifs in peptides that enhance MRI contrast). The advantage of this method for protein engineering is that virtual screening enables examination of both a larger number of variants and variants with wider sequence diversity compared to experimental directed evolution methods. The challenge of this method is to develop a model for screening the variants which correlates well with the desired functional properties of the evolved proteins. The authors choice to apply their method to discovering proteins with enhanced MRI contrast was an appropriate test of the method. A unique feature of the method is that it iterates through virtual model building, mutagenesis and screening and wet lab experiments. The winners In each round of directed evolution, or epoch, are added to the training set in order to improve the quality of the model. Screening candidates computationally drastically reduces the number of variants required for wet-lab screening. As long as POET accurately refines the model at each generation, the initial search space can be very broad, and still identify improved variants at the end of each epoch.

The algorithm was described well, but the methods and results sections need to be expanded to fully explain the experimental design and results.

Line 241 (Methods section): the Methods section should also describe the wet-lab experiments, since the results of these experiments were used to refine the models in each epoch and to evaluate the effectiveness of the algorithm. Lines 378-384 in the results describe the metric used to measure CEST contrast levels, but the acronyms MTR and RF are not defined, and it is unclear what tau refers to. While this needs clarification (and perhaps should be moved to the methods section), the rest of the experimental methods were left out. For example, how were the peptides synthesized/purified? What solution were their CEST values measured in?

Line 295: di and fi need to be defined in the RMSE formula. Also, the section title says RMSE was used to evaluate the models built from the training data. Wasn’t RMSE also used to evaluate the fitness of each protein variant in the libraries?

Line 317: At what point was the hydrophobicity threshold applied? After mutagenesis and before calculating CEST score? Perhaps this should be added to the flow chart in Fig. 2, or perhaps a more detailed flow chart of the POET screening and mutagenesis portion should be added.

Validity of the findings

Although the manuscript suggests that POET may be a valuable approach to protein engineering, some of the results are not shown or discussed clearly enough for readers to conclude that the method successfully identified proteins with higher MRI contrast.

Lines 385-387: This paragraph states that the CEST contrast value of the predicted peptides improved up to 400%. However, this cannot be evaluated, because the data is not shown. The data may be in the CEST contrast values in Table 2, but the table is not sorted by epoch. It would be useful to list the entries in Table 2 according to what epoch they were added to the data set. The CEST values (or the average value at each epoch) for the peptides in the initial data set and those added at each epoch should also be plotted to illustrate whether the CEST values increased, as expected.

Line 336 (Evolution of Models per Epoch section, Figures 5, 6, Table 4): This section needs to be reorganized and slightly expanded to explain what the different RMSE measurements are and why they go up in some cases and down in others. For example, Fig. 5 and 6, which examine two different types of RMSE, are described before Table 4, which lists what the different RMSE’s are. This made understanding why RMSE increased with epoch number in Fig. 5 but decreased in Fig. 6 very difficult to understand. It was difficult to understand why RMSE would increase at all, if the method is successful. The increase in RMSE was explained twice. The first explanation (lines 343-345) was unclear. The second explanation (lines352-354) makes more sense, but is still a little vague.

Line 349 (“Test and training fitness values show a slight over-fitting of the models”): What data or analysis supports this conclusion?

Line 350-1: The best models have 96-97 rules and express ~44 of them. For context, how many rules are used/expressed in average or poor models? Was there some expectation that maximizing the number of rules would improve the model? Why?

Additional comments

General question for discussion: if there are multiple local optima, will model-building in POET generate an “average” model that doesn’t really fit any of the optima? Perhaps this would help explain the increase in Training fitness or test fitness RMSEs as more sequences are added to the data set in each epoch. In future version of the algorithm, would it be possible or valuable to try to identify multiple optima and generate separate models for each one?

Reviewer 2 ·

Basic reporting

no comment

Experimental design

no comment

Validity of the findings

no comment

Additional comments

The authors of this manuscript present a detailed description of POET (Protein Optimization Engineering Tool), a method based on genetic programming and evolutionary computation to support the process of Directed Evolution to find proteins with better functionalities.

The authors use POET to predict proteins with better CEST levels. In addition, the experiments generated a dataset with protein motifs and their corresponding CEST contrast values.

It is relevant to mention that the authors provide the Python code and the data used in the tests, which facilitates the reproducibility of the results.

The topic, methods described, and conclusions of this article are of interest to readers of the PeerJ Physical Chemistry, Bioinformatics, and Genomics section. This article is well-organized, the writing is clear, and the statements are well justified and supported with references. The operation of the POET tool is explained in considerable detail, facilitating its understanding.

My observations are the following:
• Although an interesting description of methods is presented in the "Related Works" section, I believe a greater focus on ML-DE approaches is missing. It is also important to explicitly mention if other methods deal with the same application as this work (CEST contrast). If such methods exist, it would be interesting to compare them with POET for the selected application.
• How were the parameters used determined? (Number of runs, model population size, number of generations, number of epochs, percentage of mutations, etc.)
• It is mentioned in lines 327 and 417 that the dataset has a size of 163. However, in Table 2 there are 162 sequences (unless I made a mistake when counting).
• It is mentioned in lines 258-260 that the reverse sequence of the motif is also evaluated for each rule, but the justification is unclear to me. Could you provide more details on this?
• Alphafold is mentioned in lines 148-150. It would be interesting to add some comments regarding Alphafold2 since it is a more recent proposal that achieves better results.
• It is stated in line 59 that the search space consists of 10^20 sequences. However, it should be 20^10 sequences (20 possible values for each of the 10 positions in the sequence).
• There is a typo at the end of line 164.
• The reference in line 187 is confusing since the years do not match.
• Below line 295, the RMSE formula is presented. It would be interesting to define d_i and f_i in the context of the problem addressed.
• It is written at line 174: “Wu et al. (Wu et al., 2019) proposed an ML-DE…”. I think it is redundant to present it this way. To provide other examples, this situation is presented on lines 169, 187, 192, and 219.

---

## Round 0.2 · accepted · Accept

Thank you for carefully revising the manuscript to address the Reviewers' concerns. The current version is much improved and more accessible to readers.